# N-N-Substituted Piperazine, Cmp2, Improves Cognitive and Motor Functions in 5xFAD Mice

**DOI:** 10.3390/ijms26104591

**Published:** 2025-05-10

**Authors:** Nikita Zernov, Daria Melenteva, Viktor Ghamaryan, Ani Makichyan, Lernik Hunanyan, Elena Popugaeva

**Affiliations:** 1Laboratory of Molecular Neurodegeneration, Peter the Great St. Petersburg Polytechnic University, 195251 St. Petersburg, Russia; 2Laboratory of Structural Bioinformatics, Institute of Biomedicine and Pharmacy, Russian-Armenian University, Yerevan 0051, Armenialernik.hunanyan@rau.am (L.H.)

**Keywords:** Alzheimer’s disease, TRPC6 channel, behavior, piperazine

## Abstract

The piperazine derivative N-(2,6-difluorophenyl)-2-(4-phenylpiperazin-1-yl)propanamide (cmp2) has emerged as a potential transient receptor potential cation channel, subfamily C, member 6 (TRPC6) modulator, offering a promising pathway for Alzheimer’s disease (AD) therapy. Our recent findings identify cmp2 as a novel compound with synaptoprotective effects in primary hippocampal cultures and effective blood–brain barrier (BBB) penetration. In vivo studies demonstrate that cmp2 (10 mg/kg, intraperitoneally) restores synaptic plasticity deficits in 5xFAD mice. This study further shows cmp2’s selectivity towards tetrameric TRPC6 channel in silico. Acute administration of cmp2 is non-toxic, with no indications of chronic toxicity, and Ames testing confirms its lack of mutagenicity. Behavioral assays reveal that cmp2 improves cognitive functions in 5xFAD mice, including increased novel object recognition, better passing of the Morris water maze, and improved fear memory, as well as upregulation of motor function in beam walking tests. These findings suggest that cmp2 holds promise as a candidate for AD treatment.

## 1. Introduction

AD presents a formidable challenge in the search for effective drug therapies. A primary focus in this field has been the “amyloid-β (Aβ) cascade hypothesis”, which posits that the accumulation of Aβ peptides in the brain is the root cause of AD [1]. Since its proposal in 1992 [2], this hypothesis has shaped drug development strategies for AD, and recent years have seen notable advancements, including Food and Drug Administration (FDA) approval of two Aβ-targeting antibodies, Aduhelm and Leqembi [3]. However, they have demonstrated only modest clinical benefits and come with severe side effects, including amyloid-related imaging abnormalities (ARIAs) [4,5,6]. Notably, in 2024, Biogen withdrew Aduhelm from the market due to its unfavorable risk–benefit profile, and Leqembi failed to secure approval from the European Medicines Agency (EMA/337466/2024). This limited success underscores the urgent need for innovative AD therapies that go beyond amyloid-targeted approaches. Developing alternative treatments may hold promise for creating a more effective, safer approach to managing AD progression.

The activation of the TRPC6 presents a promising pathway for developing AD therapies [7,8]. TRPC6 can be activated by several compounds [9,10], including hyperforin, a specific and positive TRPC6 agonist that has been tested in clinical trials for mild to moderate depression [11,12]. However, hyperforin has limitations; it is unstable, challenging to synthesize [13], and exhibits side effects, including protonophore properties [14]. Due to these challenges, alternative TRPC6 agonists are needed for clinical application.

Our previous work identified a blood–brain barrier-permeable (confirmed by pharmacokinetics (PK) studies) piperazine derivative, cmp2, which binds to the central part of monomeric TRPC6 in a manner similar to hyperforin [15]. We demonstrated that cmp2 is a selective TRPC6 activator that does not enhance TRPC3 or TRPC7 activity. In AD model studies, 100 nM cmp2 successfully restored mushroom spine density in Aβ42-treated hippocampal cultures. Furthermore, 14-day intraperitoneal injections of cmp2 at a 10 mg/kg dose were shown to restore long-term potentiation (LTP) in brain slices from aged 5xFAD mice, supporting its potential as a viable therapeutic option in AD treatment.

Piperazine derivatives have been shown to inhibit both acetylcholinesterase (AChE) and butyrylcholinesterase (BuChE) [16]. Pharmacological compounds, with effects on the cholinergic system, have been used to reduce AD symptoms and improve performance in similar behavioral tasks to the ones used in the current article [17,18,19]. Indeed, our recent in silico data demonstrate the influence of cmp2 on the cholinergic system [20].

In the current study, we present a battery of behavioral testing of cmp2 at a 10 mg/kg dose. We provide data demonstrating its therapeutic effect on the cognitive and motor functions of 5xFAD mice. We further prove the selectivity of cmp2 towards tetrameric TRPC6. Our findings also demonstrate that cmp2 exhibits no acute toxicity and lacks chronic toxicity potential. Additionally, results from the Ames test confirm that cmp2 possesses no mutagenic properties. Behavioral assessments indicate that cmp2 effectively ameliorates cognitive impairments in 5xFAD mice, as evidenced by improved performance in the novel object recognition test, the Morris water maze, and the fear conditioning test. Furthermore, cmp2 enhances motor function in 5xFAD mice, as shown in the beam walking test. These findings suggest that cmp2 holds promise as a potential therapeutic agent for AD.

## 2. Results

### 2.1. Molecular Dynamic Simulation of Cmp2 with Tetrameric TRPC3 and TRPC6

Previously, using molecular docking with monomeric TRPC6, we have identified key amino acids positions that interact with cmp2 [15]. Moreover, molecular dynamic simulation has demonstrated stable interaction of cmp2 with monomer TRPC6. In vitro calcium imaging confirmed that cmp2 is a selective activator of TRPC6 over TRPC3 and TRPC7 [15].

Here, we present expanded in silico data of cmp2 complexes with tetrameric TRPC3 and TRPC6 that further prove the selectivity of cmp2 towards TRPC6. We do not present molecular dynamic simulations with tetrameric TRPC7, since there is no cryo-EM or AFM structure in the literature.

To analyze the stability of cmp2 with either tetrameric TRPC3 or TRPC6, we performed molecular dynamic simulation using previously published structures of TRPC3 (PDB ID: 7DXB) and TRPC6 (PDB ID: 5YX9). The root mean square deviation (RMSD) results for cmp2-TRPC3 complex indicate that trajectory shifts occur between 35 and 37 ns, with values around 1.970 ± 0.09 Å, and from 37 to 52 ns, the RMSD stabilizes to approximately 0.862 ± 0.04 Å (Figure 1A). Intermittent fluctuations are also present, with jumps averaging around 0.584 ± 0.03 Å. Complex cmp2-TRPC6 exhibits high stability throughout the simulation, with no significant trajectory shifts relative to the binding site (Figure 1B). The mean RMSD value is approximately 0.741 ± 0.03 Å. Thus, the cmp2 complex with TRPC3 appears less stable than that with TRPC6, as corroborated by RMSD variation calculations (TRPC6 exhibits minimal variation at 0.077, while TRPC3 has higher values at 0.497).

To further evaluate the cmp2-TRPC3 and cmp2-TRPC6 complexes, we calculated the interaction energy values for all three complexes. Our results for Coulombic and Van der Waals forces indicate no interaction of cmp2 with TRPC3 as the values for the electrostatic forces are positive, with Coulombic forces at approximately 3.89 ± 0.19 kJ/mol (Figure 1C) and Van der Waals interactions at approximately 1493.29 ± 74.65 kJ/mol (Figure 1E). However, an interaction between cmp2 and TRPC6 is present and is primarily driven by electrostatic forces, particularly Van der Waals interactions. The calculated value for Coulombic forces is approximately −17.86 ± 0.89 kJ/mol (Figure 1D). The average value of Van der Waals interactions is approximately −111.65 ± 5.58 kJ/mol (Figure 1F).

The findings suggest that cmp2 establishes a single hydrogen bond (Figure 1F) with ASN617, which enters the TRPC6 binding site 16. At the same time, it should be noted that the probability of this type of interaction does not exceed 6.54%.

Thus, obtained in silico data further confirm the selectivity of cmp2 towards tetrameric TRPC6.

### 2.2. Cmp2 Is Not Mutagenic upon AMES Testing

The assessment of in vitro mutagenicity is a crucial component of the genotoxicity testing panel in preclinical studies of new active pharmaceutical substances. To evaluate the mutagenic potential of cmp2, the Ames test was conducted using histidine-dependent strains of Salmonella typhimurium (TA98, TA1537 for frameshift mutations and TA100, TA1535 for point mutations) and a mixture of tryptophan-dependent *Escherichia coli* strains (uvrA[pKM101] (for point mutations). The test substance was evaluated at six concentrations: 12, 40, 125, 400, 1265, and 4000 µM. Strain-specific mutagens (positive control; for details, please see the Methods section) and 2% dimethyl sulfoxide (DMSO) (negative control) were used as controls. The assay was conducted both with and without metabolic activation using the rat liver microsomal S9 fraction. The inclusion of the rat liver S9 fraction allowed the examination of the test substance’s mutagenic potential after enzymatic conversion. This fraction contains enzymes like cytochrome P-450, which can activate pro-mutagens into mutagens.

The solutions of cmp2 and controls were incubated with ~107 his-/trp- bacteria of each strain for 90 min in a medium containing sufficient histidine or tryptophan to allow approximately two cycles of cell division. After incubation, the bacterial cultures were diluted in a pH indicator medium lacking histidine (for *S. typhimurium*) or tryptophan (for *E. coli*) and further incubated in 384-well microplates at 37 °C for two days. Revertant bacteria, which had undergone back mutations to histidine-/tryptophan-independence, formed colonies, leading to a drop in the pH of the medium, resulting in a color change from purple to yellow. The number of wells with revertant bacteria were counted for each concentration of the test compound and compared to the values of the negative control. Each concentration was tested in triplicate to enable statistical analysis of the data. There was a significant increase in the mean number of revertant colonies in the presence of positive controls 1 and 2 for all strains with and without S9 mix (Figure 2A–E), confirming the validity of the assay.

No revertant bacteria were observed in *Salmonella typhimurium* cultures when cmp2 was administered without metabolic activation (Figure 2A–D). Cmp2 did not show any significant effect on frameshift mutation rates (two-fold or more) in TA98 (Figure 2A) and TA1537 (Figure 2B) strains both in the absence and presence of the S9 fraction. Moreover, cmp2 was not mutagenic to TA100 (Figure 2C) and TA1535 (Figure 2D) strains with or without S9 fraction. Similarly, Cmp2 did not cause a significant increase in baseline mutagenicity or a significant decrease in mutation rate during incubation with the *E. coli* combo strain in the absence and presence of S9 fraction (Figure 2E). The findings indicate that cmp2 and its metabolites do not demonstrate mutagenic properties upon Ames testing.

### 2.3. Cmp2 Does Not Demonstrate Toxic Effects at 50 and 100 mg/kg Doses During Acute Administration

We chose 50 and 100 mg/kg doses due to the solubility properties of cmp2. To determine whether cmp2 demonstrates acute toxicity, we one-time intraperitoneally (IP) injected 2-month-old wild type (WT) (C57BL/6J) female mice with cmp2 at 50 and 100 mg/kg doses or an equivalent amount of DMSO (control). The body weight and survival of mice were tested every second day after administration (Figure 3A). No statistically significant differences in the body weight (where weights were normalized to the first day of the drug administration) and survival rate (i.e., all mice tested survived) were found between these three groups.

### 2.4. Cmp2 Does Not Demonstrate Toxic Effects at a 10 mg/kg Dose During Chronic Administration

To test the chronic toxicity of cmp2, 4-month-old WT mice were injected IP with cmp2 at a 10 mg/kg dose (cmp2 group) or an equivalent amount of DMSO (control) for 29 consecutive days. No drug-dependent chronic toxicity effect on either body weight or survival rate (i.e., all mice tested survived) was found. (Figure 3B).

We chose 5xFAD mice as an animal model that can encapsulate some hallmarks of AD; in particular, 5xFAD mice demonstrate an aggressive phenotype of AD, starting to show amyloid plaques at 2 months of age. Their cognitive functions show signs of decline from 4 to 6 months of age, and motor dysfunction can usually be observed by the time mice reach approximately 9 months of age [21]. Moreover, 5xFAD mice are often used in preclinical studies (https://alzped.nia.nih.gov/ (accessed on 15 December 2024)). Following toxicity testing, we proceeded with a series of behavioral assessments to evaluate the effect of cmp2 on the cognitive and motor functions of WT and 5xFAD mice. To this end, 8-month-old WT and 5xFAD mice were administered IP injections of cmp2 at a 10 mg/kg dose daily for 18 days before the first behavioral test. The treatment was then continued daily until the last day of behavioral testing. The schematic of behavioral tests is shown in Figure 4.

### 2.5. Effects of Cmp2 on the Cognitive and Motor Functions of WT Male Mice

Control group (WT, 6 males) of mice were intraperitoneally injected with DMSO (3.765%) diluted in saline, while “WT + cmp2” group (6 males) of mice were administered with cmp2 at a 10 mg/kg dose (cmp2 was dissolved in DMSO (3.765%) and diluted in 0.9% saline (96.235%).

To investigate the effect of IP injections of cmp2 on the recognition memory of WT mice, we employed the novel object recognition (NOR) test. Before the beginning of the experiment, mice were habituated to the arena for 10 min. On the next day, mice were placed in the middle of the arena and allowed to explore two identical objects for 10 min. Following a 24 h interval, one of the objects was substituted with a novel object, and the mice were subsequently allowed 10 min of free exploration. Exploration time was defined as the mouse’s nose being within the object zone. Exploration time and number of entries were calculated and are presented in Figure 5. In WT versus WT + cmp2 experimental groups, no significant difference in the exploration time of the novel object zone (Figure 5A) or the number of entries into the novel object zone (Figure 5B) was observed between these two groups on the second day of NOR testing.

To explore the effects of cmp2 on spatial learning and memory in WT mice, the Morris water maze test was performed after 19 days of cmp2 therapy. No significant difference in the number of successful trials during training days (Figure 5C) or time in the platform zone (Figure 5D) was observed between the WT and WT + cmp2 experimental groups.

To investigate the effect of cmp2 on the cognitive function of WT mice, a fear conditioning test was conducted after 26 days of treatment. No significant differences between WT and WT + cmp2 groups were observed in contextual memory on the 3rd or 10th day of experiment (Figure 6B,D). A significant improvement in cued fear memory was observed in the WT + cmp2 group on the 3rd day of the experiment (Figure 6C), but this effect disappeared on the 10th day of the experiment (Figure 6E).

We evaluated the impact of cmp2 on the motor function of 8-month-old WT male mice using the beam walking test. The mice received intraperitoneal injections for 33 days prior to the experiment and then every day of the experiment. On the second and third days of training, there was a statistical difference in time taken to cross beams in the WT + cmp2 group (Figure 7A,C,E). It is important to note that the number of paw slips on the 12 mm beam was statistically higher in the WT + cmp2 group on the second day of training (Figure 7D). Similar results were observed on the 8 mm beam; the number of paw slips was statistically higher in the WT + cmp2 group on the first, second, and third days of training (Figure 7F). No differences in time taken to cross the beam, paw slip numbers, or crawling scores were observed between WT and WT + cmp2 groups on day 4 of the experiment (Figure 7A–F,H), indicating that cmp2-mediated effects on motor functions in WT mice were transient and abolished by training.

### 2.6. Cmp2 Improves the Recognition Memory of 5xFAD Male Mice in the Novel Object Recognition Test

To investigate the effect of IP injections of cmp2 on the recognition memory of 5xFAD mice, we employed the novel object recognition (NOR) test. One day before the test, animals were individually habituated to the arena (Figure 8A) for 10 min. On the first day of the NOR test (after 17 days of cmp2 therapy), two identical objects (Object A and Object B) were placed in opposite sectors of the arena (Figure 8A). On the second day of testing, Object B was replaced with a novel object (Figure 8A).

All experimental groups displayed similar preferences for the identical objects on the first day of testing (Figure 8D,F, Mann–Whitney test or *t*-test; ns: non-significant.). However, on the second day, WT and 5xFAD + cmp2 mice showed a marked interest in the novel Object B compared to the familiar Object B (Figure 8E, statistical power ≥ 0.6, *: *p* < 0.05; Figure 8G, statistical power ≥ 0.7, *: *p* < 0.05) and compared to Object B from the first day (Figure 8C, statistical power ≥ 0.8, *: *p* < 0.05). In contrast, 5xFAD mice did not show a statistically significant difference in exploration time for the novel object compared to the original Object B from the first day (Figure 8C, ns: non-significant). Additionally, no significant difference in the number of entries into the novel object zone was observed in 5xFAD mice on the second day (Figure 8E). Notably, 5xFAD mice demonstrated a significantly reduced preference for the novel object compared to 5xFAD + cmp2 mice (Figure 8C,G, one-way ANOVA following Dunnett’s multiple comparisons test between 5xFAD + cmp2 and the other treatment groups. **: *p* < 0.01, ns: non-significant).

### 2.7. Cmp2 Improves the Spatial Memory of 5xFAD Mice in the Morris Water Maze Task

To explore the effects of cmp2 on spatial learning and memory, the Morris water maze test was performed after 19 days of cmp2 therapy (Figure 9). During the Morris water maze test, the movement trajectories of the mice in each group were registered (Figure 9A). The percentage of successful attempts was calculated for each group during the 5-day training period (Figure 9B). The learning curves all groups showed an upward trend, indicating that all groups of mice were taught to find the escape platform (Figure 9B day 1 vs. day 5). Among all, the 5xFAD group showed a statistically marked retardation in the percentage of successful trials compared with the WT group on the last day of training (Figure 9B day 5). However, on the 5th day of training, there was no significant difference in this criterion between the WT and 5xFAD + cmp2 groups (Figure 9B, WT + vehicle, mean ± SEM, 96.87% ± 1.10%, 5xFAD + vehicle, 64.50% ± 3.17%, 5xFAD + cmp2, 69.44% ± 2.70%, Kruskal–Wallis test with Dunn’s post hoc, *: *p* < 0.05, ns: non-significant, statistical power = 0.8).

During the probe day trial, the number of crossings to the platform location (Figure 9C), the latency of the first entry to the platform location (Figure 9D), and the time spent in the target quadrant (Figure 9E) were analyzed. The number of platform crossings was significantly reduced in the 5xFAD group compared with the WT (Figure 9C). However, no significant difference in the number of platform crossings was observed between the WT and 5xFAD + cmp2 groups (Figure 9C, WT + vehicle, mean ± SEM, 2.25 ± 0.45, 5xFAD + vehicle, 1.00 ± 0.47, 5xFAD + cmp2, 1.44 ± 0.38, Kruskal–Wallis test with Dunn’s post hoc, *: *p* < 0.05, ns: non-significant, statistical power = 0.4). A similar pattern was observed for the latency to the first entry to the platform location (Figure 9D, WT + vehicle, mean ± SEM, 23.55 s ± 6.07 s, 5xFAD + vehicle, 63.51 s ± 10.96 s, 5xFAD + cmp2, 49.08 s ± 7.35 s, one-way ANOVA following Dunnett’s multiple comparisons test, **: *p* < 0.01, ns: non-significant, statistical power = 0.8) and the time spent in the target quadrant on the probe day (Figure 9E, WT + vehicle, mean ± SEM, 19.94 s ± 1.96 s, 5xFAD +vehicle, 11.52 s ± 2.43 s, 5xFAD + cmp2, 19.43 s ± 2.40 s, one-way ANOVA following Dunnett’s multiple comparisons test, *: *p* < 0.05, ns: non-significant, statistical power = 0.7).

Taken together, the obtained results indicate that 20-day-long therapy with cmp2 at a 10 mg/kg dose has the potential to alleviate spatial memory deficits in 8-month-old 5xFAD mice.

### 2.8. Cmp2 Recovers the Deficit of Contextual and Cued Memory in 8-Month-Old 5xFAD Mice

To investigate the effect of cmp2 on the cognitive function of diseased mice, a fear conditioning test was conducted in 5xFAD mice. Two cohorts of 5xFAD mice, comprising 9 and 10 mice, respectively, received intraperitoneal injections of either cmp2 (10 mg/kg) or an equivalent volume of vehicle, which were administered for 25 days prior to the experiment and continued daily throughout the experimental period. Additionally, a control cohort of wild-type (WT) mice (n = 9) was administered an equivalent volume of vehicle.

All mice demonstrated an increase in freezing behavior on the testing day compared to training day 3 (Figure 10B,C), indicating that all groups of mice were well trained by the contextual fear-conditioning process (Figure 10B, training vs. context: WT + vehicle, 0.96% ± 1.72% vs. 19.54 ± 16.15% (statistical power = 0.91); 5xFAD + vehicle, 10.12% ± 14.58% vs. 24.54% ± 11.46% (statistical power = 0.62); 5xFAD + cmp2, 3.51% ± 5.78% vs. 11.23% ± 10.85% (statistical power = 0.43); Mann–Whitney test, ***: *p* < 0.001, *: *p* < 0.05) and the cued fear-conditioning process (Figure 10C, pre-tone vs. tone: WT + vehicle, 5.65% ± 3.22% vs. 43.45 ± 17.09% (statistical power = 0.99); 5xFAD + vehicle, 18.61% ± 16.09% vs. 40.73% ± 27.70% (statistical power = 0.52); 5xFAD + cmp2, 25.81% ± 17.89% vs. 59.17% ± 31.18% (statistical power = 0.72); *t*-test, ***: *p* < 0.001, *: *p* < 0.05). However, no significant differences in freezing time were observed between training day and context day 10 in the untreated 5xFAD group (Figure 10D, training vs. context: 5xFAD + vehicle, 10.12% ± 14.38% vs. 6.69% ± 4.81%; Mann–Whitney test, ns: non-significant), indicating that 5xFAD mice had impaired contextual memory. In contrast, significant differences between training and context days persisted in the WT and cmp2-treated 5xFAD groups (Figure 10D training vs. context: WT + vehicle, 0.96% ± 1.72% vs. 11.23 ± 10.87% (statistical power = 0.8); 5xFAD + cmp2, 3.52% ± 5.79% vs. 23.41% ± 18.35% (statistical power = 0.83); Mann–Whitney test, **: *p* < 0.01), confirming that cmp2 can partially rescue contextual memory. Furthermore, on day 10 of the experiment, 5xFAD mice exhibited a significant contextual fear memory deficit relative to their cmp2-treated counterparts (Figure 10D, Kruskal–Wallis test following Dunn’s multiple comparisons test between 5xFAD + cmp2 and the other groups of treatment. *: *p* < 0.05). A similar trend of cmp2-mediated recovery of cued fear in 5xFAD-treated mice was observed on day 10 of the cued conditioning test (Figure 10E pre-tone vs. tone: WT + vehicle, 4.79% ± 1.50% vs. 30.46 ± 18.67% (statistical power = 0.96); 5xFAD + vehicle, 15.07% ± 12.47% vs. 29.47% ± 19.41%; 5xFAD + cmp2, 17.37% ± 12.86% vs. 53.63% ± 31.03% (statistical power = 0.86); *t*-test or Mann–Whitney test, **: *p* < 0.01, *: *p* < 0.05).

The obtained results demonstrate that 26-day-long therapy with cmp2 at a 10 mg/kg dose can alleviate contextual and cued memory deficit in 8-month-old 5xFAD mice.

### 2.9. Cmp2 Recovers the Motor Performance of 8-Month-Old 5xFAD Mice

Just like in AD patients, 5xFAD mice display several motor impairments [22]. Transgenic mice overexpressing TRPC6 demonstrate enhanced motor performance following ischemia in the rotarod test [23]. Another study showed a reduction in TRPC6 expression following spinal cord injury. Following the administration of HYP9 (a TRPC6-specific agonist), an increase in TRPC6 expression, a reduction in the number of active astrocytes, and the restoration of motor function in the limbs were observed [24]. In the current study, we evaluated the impact of cmp2 on the motor function of 8-month-old mice using the beam walking test. The mice received intraperitoneal injections for 33 days prior to the experiment and then every day of the experiment (Figure 3). The “5xFAD + cmp2” group received cmp2 injections at a concentration of 10 mg/kg, while the control groups, 5xFAD and WT, received DMSO injections in equivalent volumes.

On day 34 of the injection period, the mice underwent a 4-day training session for the beam walking test. Differences in locomotor functions between the wild-type mice and 5xFAD mice were observed. The 5xFAD mice took significantly more time to traverse beams of varying thicknesses compared to the wild-type mice (Figure 11A,C,E, statistical power = 0.99). Additionally, the 5xFAD mice exhibited difficulties in traversing the beams (Figure 11H, statistical power = 0.99), predominantly adopting a “crawling” motion (walking score = 0, Figure 11G).

Cmp2 injections resulted in a significant reduction in the time required to cross the beams compared to the 5xFAD group (Figure 11A (statistical power = 0.8), C (statistical power = 0.9), E (statistical power = 0.7)). The mean crossing time on the 18 mm beam was 14.8 s for the cmp2-injected 5xFAD mice versus 18.9 s for the 5xFAD group; on the 12 mm beam, the time was 17.3 s versus 21.1 s, respectively; and on the 8 mm beam, 19.9 s versus 22.5 s, respectively. Improvement in the locomotion of the 5xFAD + cmp2 group was also evident on the 8 mm beam, as measured by the “number of slips” parameter (Figure 11F). When assessing “crawling” vs. normal movement, statistically significant differences between the 5xFAD and 5xFAD + cmp2 groups were observed only on the 8 mm beam, where the mean score for the cmp2-injected mice was 25 compared to 3.3 for the 5xFAD group (Figure 11H, statistical power = 0.81).

Taken together, the obtained findings indicate that on the 4th day of the beam walking test, cmp2 could alleviate motor impairment in 8-month-old 5xFAD mice.

### 2.10. Cmp2 Intraperitoneal Injections for 38 Days Do Not Influence Amyloidosis or Astrogliosis in the Mouse Hippocampus

This study evaluated the influence of cmp2 on amyloid plaque accumulation and the activation state of astroglia in the hippocampus of 9-month-old mice. Following behavioral assessments, fixed brain slices from three experimental groups were prepared for immunohistochemical analysis. The presence of multiple amyloid plaques in the hippocampi of 5xFAD mice was confirmed, with a measured amyloid plaque area of 1.56 ± 0.43% in 5xFAD mice compared to 0.045 ± 0.036% in wild-type (WT) mice (Figure 12B). Similarly, the number of plaques per unit area was significantly higher in 5xFAD mice (84 ± 16 plaques/mm^2^) compared to WT mice (6 ± 6 plaques/mm^2^) (Figure 12B). Cmp2 injections led to a reduction in both the amyloid plaque area and plaque number in the hippocampi of 5xFAD mice. However, these reductions were not statistically significant compared to the untreated 5xFAD group (Figure 12B). Specifically, in 5xFAD + cmp2 mice, the amyloid plaque area was 1.14 ± 0.46%, and plaque count was 74 ± 14 plaques/mm^2^.

Immunohistochemical analysis confirmed significant astrogliosis in the hippocampus of 9-month-old 5xFAD mice (Figure 13). Astroglia were stained with glial fibrillary acidic protein (GFAP) antibody. Quantification of the mean fluorescence intensity of GFAP-positive cells revealed elevated levels of active astroglia in 5xFAD mice (mean grey value: 1087 ± 302) compared to wild-type (WT) mice (mean grey value: 915 ± 266) (Figure 13B). Similarly, the number of GFAP-positive cells per mm^2^ was significantly higher in 5xFAD mice (247 ± 54) compared to WT mice (184 ± 57) (Figure 13B).

Administration of cmp2 at a dose of 10 mg/kg for 38 days did not result in a statistically significant reduction in the number of active astroglia cells in the hippocampi of 5xFAD mice when compared to the untreated 5xFAD group (Figure 13B). In the 5xFAD + cmp2 group, the mean fluorescence intensity of GFAP was 1036 ± 266, and the number of GFAP-positive cells per mm^2^ was 251 ± 55.

## 3. Discussion

The findings presented build on previous research that explored the synaptoprotective effects of cmp2 [15], a selective TRPC6 positive modulator, in a 5xFAD mouse model of AD. This model is widely used for studying AD as it exhibits hallmark AD pathologies, including amyloid-beta accumulation, synaptic deficits, and cognitive decline [21]. The prior work demonstrated that cmp2 could effectively restore synaptic plasticity deficits in these mice, an encouraging result given that synaptic dysfunction and loss are strongly correlated with cognitive impairment in AD patients [7].

The results of molecular dynamics simulations indicate that the cmp2 interaction exerts a directional effect on TRPC6, as there is no affinity for TRPC3. This is corroborated by obtained energy parameters of the complexes. Furthermore, cmp2 forms one hydrogen bond with one of the key amino acids included in the TRPC6 binding site.

In the current study, cmp2 was further evaluated for its safety profile and therapeutic potential in treating cognitive and motor impairments associated with AD. Acute administration of cmp2 was shown to be non-toxic (Figure 3A) without any signs of chronic toxicity (Figure 3B), as supported by the Ames test (Figure 2), which confirmed that cmp2 does not exhibit mutagenic properties. Thus, cmp2’s safety profile suggests that it could be a feasible candidate for long-term treatment approaches.

Behavioral studies demonstrate that cmp2 (10 mg/kg, intraperitoneal) has differential effects on cognitive and motor functions in WT and 5xFAD mice.

No significant impact on recognition memory (NOR, Figure 5), spatial learning (Morris water maze, Figure 5), or contextual fear memory (Figure 6) was observed in WT mice treated with cmp2. Transient improvement in cued fear memory was present in WT + cmp2 group on day 3 but, disappeared by day 10 (Figure 6). Cmp2 influenced motor function in WT mice (shown by decreased beam-crossing time and increased paw slips) during early training days, but there were no lasting effects by day 4, suggesting adaptation.

Regarding cmp2 rescue of cognitive deficits in 5xFAD mice, we observed improved novel object recognition, restored exploration preference similar to WT mice (Figure 8), and enhanced spatial memory in the Morris water maze, with 5xFAD + cmp2 mice performing comparably to WT upon probe day (Figure 9). We observed restored contextual and cued fear memory, with cmp2-treated 5xFAD mice showing significant freezing behavior compared to untreated 5xFAD mice (Figure 10), as well as recovered motor functions. Cmp2 reduced beam crossing time and the number of paw slips, particularly on narrower beams (8 mm); it also improved movement strategy, with fewer “crawling” episodes, indicating better coordination (Figure 11).

The overall implication is that cmp2 has limited effects in healthy WT mice, with only transient changes in cued memory and motor function. In 5xFAD mice, cmp2 effectively mitigates AD-related deficits in recognition, spatial, and fear memory, while also improving motor coordination. This study highlights cmp2 as a promising candidate for alleviating cognitive and motor dysfunction in neurodegenerative conditions, particularly in AD models, warranting further investigation into its neuroprotective mechanisms.

Interestingly, immunohistochemistry (IHC) analysis revealed that cmp2 did not influence amyloidosis (the buildup of amyloid-beta plaques) or astrogliosis (increased activity of astroglia) (Figure 12 and Figure 13). This finding is notable because it suggests that cmp2 is able to improve cognition and motor function even in the presence of amyloid and astrogliosis pathology, indicating that cmp2 treatment might work at advanced stages of the disease.

Given that amyloid-targeting therapies have shown limited clinical success [4,5,6,25], cmp2’s mechanism of action may provide a distinct advantage by addressing AD through a pathway that bypasses direct amyloid interaction. Instead, cmp2’s activation of TRPC6 channels could support neuronal function and synaptic integrity, offering neuroprotection in a way that is less reliant on reducing amyloid-beta accumulation.

It is important to note that our previous in silico data demonstrate the influence of cmp2 on the cholinergic system [20]. Thus, the likelihood of cmp2 having a positive impact on 5xFAD animal behavior via modulating TRPC6 and cholinergic-dependent signaling pathways is high. However, cmp2’s effect on the cholinergic system has been confirmed only in silico. Further in vitro and in vivo experiments are needed to validate cmp2 as an effective AChE and BuChE inhibitor.

In summary, the current research strengthens the case for cmp2 as a promising form of AD therapy. While cmp2 demonstrated robust improvements in cognitive and motor performance in 5xFAD mice, we observed no significant reduction in amyloid plaque burden or astrogliosis. This intriguing dissociation suggests cmp2 may primarily exert symptomatic rather than disease-modifying effects in this model, though we cannot exclude subtle modifications of soluble oligomers or synaptic pathology not captured by our measurements. Several non-exclusive mechanisms could account for these functional benefits, detailed as follows.

TRPC6-mediated neuroplasticity: As a TRPC6 activator, cmp2 may enhance synaptic resilience independent of amyloid clearance. TRPC6 channels regulate dendritic spine formation via Ca^2^⁺-dependent CREB phosphorylation [26], potentially compensating for existing synaptic deficits.

Downstream signaling modulation: TRPC6 activation stimulates BDNF/TrkB pathways [27], which could restore neuronal function despite persistent plaques.

Cholinergic involvement: TRPC6 might regulate excitation of cholinergic neurons [28]. Cmp2 might augment acetylcholine release, mitigating functional impairment, similar to donepezil’s symptomatic benefits.

These mechanisms emphasize that functional recovery in neurodegenerative models need not require plaque clearance. However, longitudinal studies with earlier cmp2 intervention are needed to definitively exclude disease-modifying effects on Aβ dynamics. Taken together, these findings highlight TRPC6 activation as a promising symptomatic strategy for slowing down AD progression.

### Limitations of the Study

(A)Testing only a single dose (10 mg/kg) represents a limitation of the current study. Comprehensive PK profiling and dose–response studies would provide valuable insights into the compound’s efficacy window and target specificity. To establish optimal dosing parameters, future PK/pharmacodynamic (PD) studies with multiple dose levels and using both sexes of mice are needed.(B)Potential sex-dependent toxicity was not assessed in this study; future work should evaluate both male and female mice to identify any sex-biased effects.(C)While our study identified statistically significant differences in behavioral tests between groups, we acknowledge that the statistical power for some comparisons depicted in Figure 5, Figure 6, Figure 7, Figure 8, Figure 9 and Figure 10 did not reach the conventional threshold of 80% (0.8), particularly for effects of smaller magnitude. This limitation stems from the moderate sample sizes, which may have been insufficient to reliably detect subtle behavioral changes. Future studies with larger cohorts would help clarify the robustness of both positive and negative findings.

## 4. Materials and Methods

### 4.1. Three-Dimensional Molecular Models of Target Proteins

TRPC6 is represented by several molecular models in the RCSB database (https://www.rcsb.org/ (accessed on 15 December 2024)), although not all these models contain the full 3D structure of the target. For example, PDB ID: 6CV9 represents the cytoplasmic domain of mouse TRPC6. The human TRPC6 protein is available in six models with different ligands and resolutions. In our study, the human TRPC6 model, PDB ID: 5YX9, was determined by electron microscopy at a resolution of 3.8 Å. For TRPC3, ten human-type models are stored in the same database. We selected PDB ID: 7DXB with a resolution of 2.70 Å for our studies.

### 4.2. Molecular Dynamics

Molecular dynamics simulations were conducted using GROMACS 2020 modified (version 2.1), employing the CHARMM 36 force field, which is a standard approach for investigating protein–ligand interactions. A cubic box was selected as the virtual space, and TIP3P water was employed for solvation purposes. The desired physiological conditions were achieved by the addition of sodium and chloride ions, which served to neutralize the system. Energy minimization was performed to avoid steric clashes and unfavorable interactions. System equilibrium was reached using NVT (constant number of particles, volume, and temperature) and NPT (constant number of particles, pressure, and temperature) conditions at 300 K and 1 atm, maintained with a Nosé–Hoover thermostat and Parrinello–Rahman barostat [29]. Electrostatic interactions were modeled with the Particle-Mesh Ewald (PME) method, and bond constraints were applied using the LINCS algorithm [30]. Trajectory analysis was conducted with VMD [31] and Xmgrace (available at: http://plasma-gate.weizmann.ac.il/Grace/ (accessed on 15 December 2024)). The stability of protein–ligand complexes was evaluated by calculating RMSD and the number of hydrogen bonds. The simulation ran for 100 ns with a 2 fs time step, and cutoffs were set at 3.60 Å for hydrogen bonds, 9.00 Å for Coulombic interactions, and 14.00 Å for Van der Waals interactions. Simulations were performed on the Xeon 2.60 GHz cluster with 28 processors and 56 computational nodes at the Supercomputing Center of Peter the Great St. Petersburg Polytechnic University (www.spbstu.ru (accessed on 15 December 2024)) and the Armenian National Supercomputing Center after Sh.Aznavour (https://anscc.sci.am (accessed on 15 December 2024)), with 284 nodes. Linux OS was used for all simulations.

### 4.3. Animals

Briefly, 5xFAD mice (Jackson Laboratory, Bar Harbor, ME, USA; MMRRC Strain #034848-JAX) in a C57BL/6J background were used for behavior studies. Wild-type (WT) mice (C57BL/6J) were used for toxicity studies. Animals were housed under controlled standard conditions given with 12:12 h light–dark cycle and given ad libitum access to food and water. Initial toxicity screening was conducted in female mice in accordance with standard protocols for detecting sensitive toxicological responses. Behavioral testing was performed in male mice to avoid potential variability introduced by the estrous cycle. All studies corresponded to the principles of the Declaration of Helsinki and were approved by the Bioethics Committee of Peter the Great St. Petersburg Polytechnic University in St. Petersburg, Russia.

### 4.4. Chemical Compounds

N-(2,6-difluorophenyl)-2-(4-phenylpiperazin-1-yl)propanamide (cmp2) was synthetized by ChemRAR (Moscow, Russia).

Positive controls for the Ames test, 2-nitrofluorene (PPC-NF00), 4-nitroquinoline-N-oxide (PPC-NQ02), N4-Aminocytidine (PPC-AC02), 9-aminoacridine (PPC-AR05), 2-aminoanthracene (PPC-AA02), and 2-aminofluorene (PPC-AF10), were purchased from Xenometrix (Allschwil, Switzerland).

### 4.5. Ames Assay

This study was conducted using the plate-based Ames test to evaluate mutagenicity in histidine-dependent strains of *Salmonella typhimurium* (TA98, TA100, TA1535, TA1537) and tryptophan-dependent strains of *Escherichia coli* (combo strain uvrA[pKM101]), both with and without metabolic activation by the S9 fraction.

The test substances were examined across six concentrations: 12, 40, 125, 400, 1265, and 4000 µM. Solutions were incubated with approximately 10^7^ his-/trp- bacteria of each strain for 90 min in a medium containing sufficient histidine or tryptophan to allow for approximately two cycles of cell division. Positive controls, consisting of specific mutagens for each strain, and a negative control (2% DMSO solvent) were also employed. After incubation, the bacterial cultures were diluted in a pH indicator medium devoid of histidine or tryptophan and subsequently incubated in 384-well microplates at 37 °C for 48 h. Bacterial cells that underwent reverse mutations to histidine- or tryptophan-independent forms produced colonies, wherein a pH-driven color change from purple to yellow was observed. The number of wells containing revertant bacteria was quantified for each concentration and compared against negative control values. Experiment was repeated three times.

Positive controls, as recommended by the manufacturer and specific to each strain, included 2-nitrofluorene [2 µg/mL] (*S. typhimurium* TA98), 4-nitroquinoline-N-oxide [0.1 µg/mL] (*S. typhimurium* TA100), N4-Aminocytidine [100 µg/mL] (*S. typhimurium* TA1535), 9-aminoacridine [15 µg/mL] (*S. typhimurium* TA1537), and 4-nitroquinoline-N-oxide [2 µg/mL] (*E. coli* combo). In experiments involving metabolic activation by the S9 fraction, 2-aminoanthracene [5 µg/mL] served as the positive control for all *S. typhimurium* strains, while 2-aminofluorene [400 µg/mL] was used for the *E. coli* combo strain.

### 4.6. Chronic Toxicity Test

WT (C57BL/6J) female mice were randomly divided into two groups of 8 mice each. The assigned mice were injected IP with cmp2 (10 mg/kg diluted in saline buffer containing 10% Kleptose (parenteral-grade, pyrogen-free hydroxypropyl β-cyclodextrin [HPβCD], Roquette Frères, Lestrem, France, CAS No. 128446-35-5) or vehicle (equivalent volume of DMSO in saline with 10% Kleptose) for 29 consecutive days. The body weight of every mouse was measured every two days.

### 4.7. The Novel Object Recognition Test

The novel object recognition test is a widely employed behavioral assay for assessing recognition memory in murine models. During the trial, the movements of the mice were recorded by a camera (Webcam Logitech C270, Apples, Switzerland) positioned above an arena. The arena consisted of an opaque, plexiglass chamber with a diameter of 62.4 cm. Before the beginning of the experiment, mice were habituated to the arena for 10 min. On the next day, mice were placed in the middle of the arena and allowed to explore two identical objects for 10 min. Following a 24 h interval, one of these objects was substituted with a novel object, and the mice were subsequently allowed 10 min of free exploration. Exploration time was defined as the mouse’s nose being within the object zone. Exploration time and number of entries were calculated using ANY-Maze tracking software version 7.49 (Stoelting Europe, Dublin, Ireland).

### 4.8. Morris Water Maze

We conducted an evaluation of learning and memory impairments in 5xFAD mice, including eight-month-old 5xFAD specimens, using the Morris water maze (MWM) test. For this assessment, the mice were introduced to a circular pool (150 cm in diameter, 66 cm in height, with water maintained at 21–23 °C) surrounded by four visual cues placed at randomly assigned locations. The pool was filled with tap water made opaque by the addition of dry milk. A circular platform (10 cm in diameter, 27.5 cm in height) was submerged 1–1.5 cm below the water surface, serving as the escape platform. Over a period of 5 days, the mice underwent training to locate the hidden platform, with 4 trials conducted per day and a minimum inter-trial interval of 20 min. The starting position of each mouse was randomized across four cardinal directions, with the sequence remaining consistent for all mice throughout the training sessions. Each trial concluded either when the mouse successfully located the platform or after 90 s, at which point mice that failed to find the platform were gently guided to it. Mice were conditioned to remain on the platform for 15 s before being removed from the pool. Mice that did not locate the platform within the trial duration were placed on the platform for 15 s, held by the tail. On the sixth day, during the probe trial phase, the platform was removed, and mice were allowed to swim freely for 90 s. The performance of each mouse during the trials was meticulously recorded using ANY-maze automated behavioral video tracking software, version: 7.49 (Stoelting Europe, Dublin, Ireland).

### 4.9. Fear Conditioning Test

Eight-month-old 5xFAD and wild-type (WT) mice were administered intraperitoneal injections of cmp2 (10 mg/kg body weight, dissolved in DMSO (3.765%) and diluted in 0.9% saline (96.235%)) or an equivalent volume of DMSO diluted in saline (vehicle control) once daily over a 24-day period. To evaluate the impact of cmp2 on the context and cued memory, the mice were subjected to a fear conditioning test (Figure 7A).

The conditioned stimulus was presented concurrently with and overlapped by the unconditioned stimulus. The experiments were conducted within a standard conditioning chamber (25 cm × 20 cm × 30 cm) equipped with a stainless-steel floor connected to an electric shock generator. This chamber was placed within a soundproof isolation cubicle. Behavioral measurements were recorded using two cameras (Webcam Logitech C270 and Webcam Logitech C200, Apples, Switzerland) connected to a computer running video freeze software (AUTO_URAI_4, version 1, developed by Dr. V.V. Sizov, Institute of Experimental Medicine, Saint Petersburg, Russia, I.P. Pavlov Department of Physiology, sizoff@list.ru).

To eliminate potential freezing behavior due to the novelty of the environment, the mice were acclimated to the chamber prior to conditioning. On day 1 (habituation), mice were individually introduced into the habituation room and placed in the training chamber for 5 min without any auditory or tactile stimuli.

On day 2 (training), each mouse was placed in the conditioning chamber for 2 min without the delivery of shocks or tones. Subsequently, the mice were exposed to three tone–shock pairings (55 dB) that coincided with a 2 s foot shock (0.3 mA, unconditioned stimulus) administered with 60 s intervals between trials. Following the last shock, the mice were allowed to recover for an additional 60 s before being returned to their home cages.

On day 3 and day 10 (testing), the mice were reintroduced to the familiar conditioning chamber, now devoid of both tones and shocks, and freezing behavior was recorded for 180 s to assess contextual memory retrieval. For the tone fear retrieval trial, conducted one hour after the contextual test, the mice were placed in a modified conditioning chamber featuring black and white striped walls, a covered floor, and a vanilla extract scent. After 180 s of baseline recording in the modified chamber, the tone used during fear conditioning training was presented for an additional 180 s, and freezing behavior was assessed both before and during the tone presentation.

Freezing time was calculated using ANY-Maze tracking software version 7.49 (Stoelting Europe, Dublin, Ireland).

### 4.10. Beam Walking Test

The beam walking test apparatus comprised 1-m-long circular beams of varying widths—18 mm, 12 mm, and 6 mm—all mounted on stands 50 cm above the floor. A platform was situated at the end of each beam, serving as the finishing point.

During the training phase, eight-month-old 5xFAD and wild-type (WT) mice underwent three trials on each beam (totaling nine trials per day) across four consecutive training days. In each trial, the mice were placed at one end of the beam, and the time taken to traverse the 1 m distance to the platform at the opposite end was recorded. The stopwatch was paused whenever a mouse ceased movement and resumed when the mouse began moving again. Additionally, the frequency of hind paw slips from the beam was documented.

To quantitatively assess beam walking performance, a 1–100 scoring system was employed. This system evaluates the degree to which the mice can traverse the beam using their paws. Each trial was assigned a score of 0, 50, or 100. Mice exhibiting significant difficulty traversing the beam, such as gripping it with all paws and crawling (dragging the affected hind limbs), were assigned a score of “0”. Mice able to traverse the beam normally, with both affected paws in contact with the horizontal surface of the beam, were assigned a score of “100”. A score of “50” was given when the mice displayed a combination of both behaviors, crossing part of the beam normally and part by crawling (Figure 8G). Thus, higher scores indicated more proficient beam walking ability.

WT mice typically traversed the beam with minimal pauses. However, should a mouse stop, sniff, or survey its surroundings without progressing, the researcher was advised to gently encourage forward movement by nudging the mouse with gloved fingers. Upon reaching the platform, the mice were allowed approximately 20 s to rest before the next trial. Following each training session, the mice were returned to their respective cages. It is important to note that overtraining or excessive familiarity with the task and apparatus can result in an increased frequency of stops. After each mouse completed the test, the beams and platform were cleaned with 70% ethanol to ensure proper hygiene.

### 4.11. Immunohistochemistry and Thioflavin Staining

After the behavioral experiments, the mice were euthanized via intraperitoneal (i.p.) injection of approximately 400 μL of urethane solution. Their brain tissues were perfused with 4% paraformaldehyde (PFA). The entire brain was then removed and fixed in PFA for 24 h at 4°C before being sectioned into 50 μm thick slices. The slices were washed for 10 min in PBS buffer.

For immunohistochemistry (IHC), antigen retrieval was performed by incubating the slices in 2N HCl for 15 min at room temperature (RT). The slices were then blocked with a solution containing 10% bovine serum albumin and 0.25% Triton X-100 for 1 h at RT. After blocking, the tissue slices were incubated overnight at 4 °C with anti-GFAP antibodies (1:1000 dilution in TBST; BioLegend, San Diego, CA, USA, Cat. 840001).

The slices were then washed three times in PBS and incubated with AlexaFluor 594 anti-rabbit antibodies (1:1000 dilution in TBST; Thermo Fisher Scientific, Waltham, MA, USA) for 1 h at RT. For thioflavin staining, the slices were incubated in a 1% aqueous solution of thioflavin T (Sigma-Aldrich, St. Louis, MO, USA) for 3 min. To reduce background fluorescence, the sections were first washed with a 1% aqueous solution of acetic acid followed by three washes in PBS.

Finally, the slices were mounted and prepared for confocal imaging. Images were obtained at 4× magnification using a Thorlabs confocal microscopy system.

### 4.12. Statistics

Data are presented as the mean ± SEM or as the mean ± SD (indicated in the text). Statistical analyses were performed using Graphpad Prism software (version 10). Sample distributions were assessed for normality (Shapiro–Wilk test) and homogeneity (Bartlett’s test). Statistical analysis was performed using the Mann–Whitney test (non-normal distribution) or *t*-test (normal distribution) between two groups. Multiple comparison analysis was performed using a one-way ANOVA (normal and homogeneous distribution) or Welch’s ANOVA (normal and non-homogeneous distribution) following Dunnett’s post hoc and Kruskal–Wallis tests (non-normal distribution) with Dunn’s post hoc. We provided a power analysis of the behavioral experiments using G* Power software, version 3.1.9.6 (retrospectively, based on effect sizes observed in our data).

## Figures and Tables

**Figure 1 ijms-26-04591-f001:**
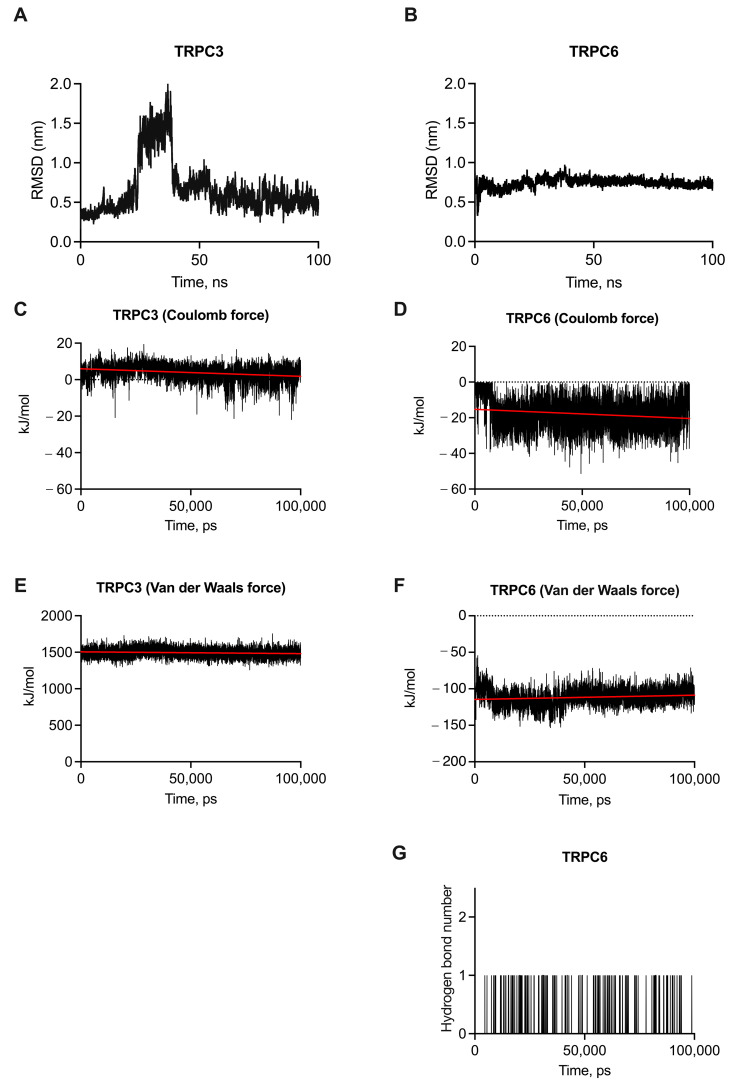
Results of molecular dynamic simulations performed for cmp2 complexes with tetrameric TRPC3 and TRPC6. (**A**,**B**) RMSD values obtained from the complex formation simulations for two targets (TRPC3 and TRPC6) with cmp2 over a 100 ns duration. Calculated interaction energies of cmp2 with TRPC3 and TRPC6 complexes: (**C**,**D**) Coulombic, (**E**,**F**) Van der Waals, and (**G**) hydrogen interactions. The red line is a a linear trend line for average value of the obtained energy.

**Figure 2 ijms-26-04591-f002:**
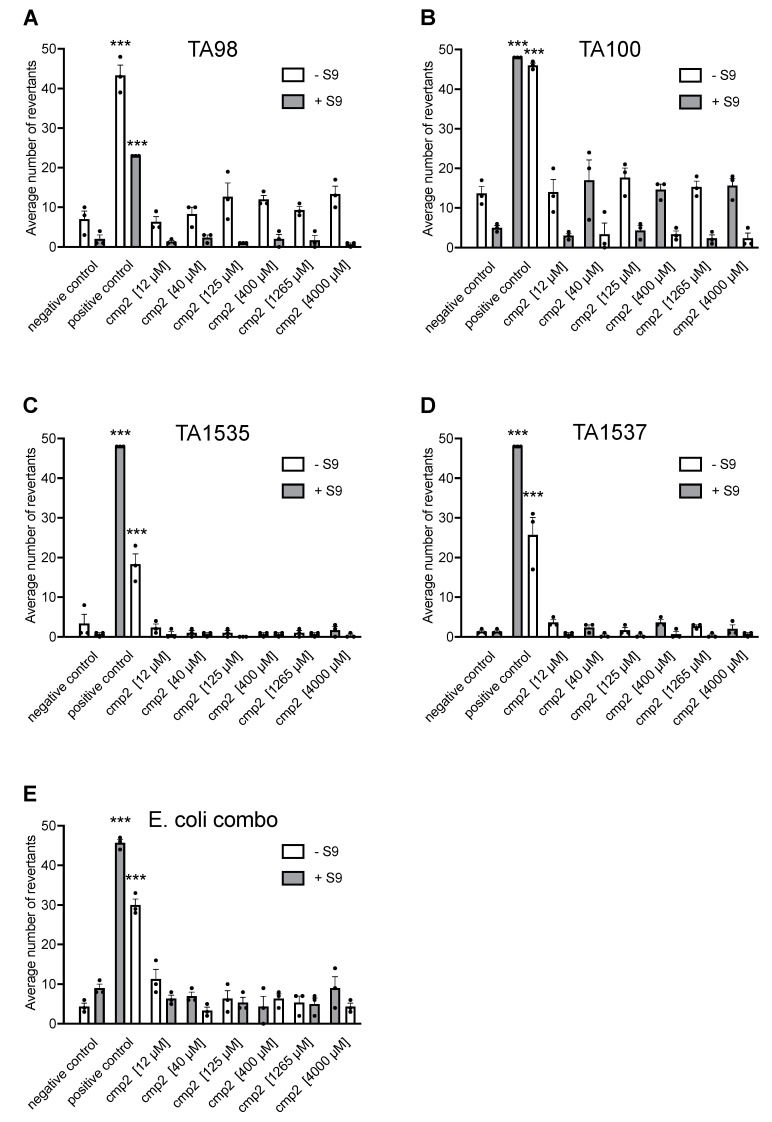
Cmp2 and its metabolites are non-mutagenic upon Ames testing. The mutagenic properties of cmp2 were investigated in the absence (white bars) and presence (grey bars) of S9 liver extract in histidine-dependent *S. typhimurium* strains (**A**) TA98, (**B**) TA1537, (**C**) TA100, (**D**) TA1535 and in a tryptophan-dependent *E. coli* strain (**E**). Data are presented as mean with SEM, individual data points are presented as dots. Experiment was repeated three times (n = 3). Statistical analysis was performed using the one-way ANOVA test with Dunnett’s multiple comparisons test between negative control and the other groups. ***: *p* < 0.001.

**Figure 3 ijms-26-04591-f003:**
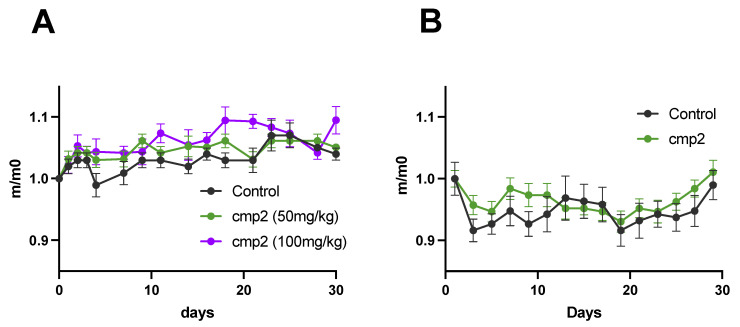
No significant effect on the body weight of mice was found after acute (**A**) or chronic (**B**) administration of cmp2. n (mice) = 5 (for acute toxicity test) and 8 (for chronic toxicity test). m/mo—weights were normalized to the first day of the drug administration. All data represent the mean ± SEM.

**Figure 4 ijms-26-04591-f004:**
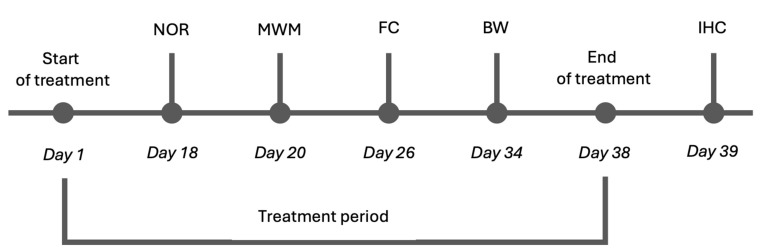
Scheme of the experiment involving male mice that were injected daily with either cmp2 at a 10 mg/kg dose or vehicle. The mice were 7.5 months old at the beginning of the experiment. The same groups of mice were used for all tests. Each behavioral test was performed at a different time point. The novel object recognition (NOR) test was performed on day 18 after first injection of cmp2; the Morris water maze (MWM) on day 20; the fear conditioning test (FC) on day 26; and the beam walking (BW) test on day 34. After the behavioral experiments, the mice were euthanized (day 39), and immunohistochemical analysis of amyloidosis and astrogliosis were performed.

**Figure 5 ijms-26-04591-f005:**
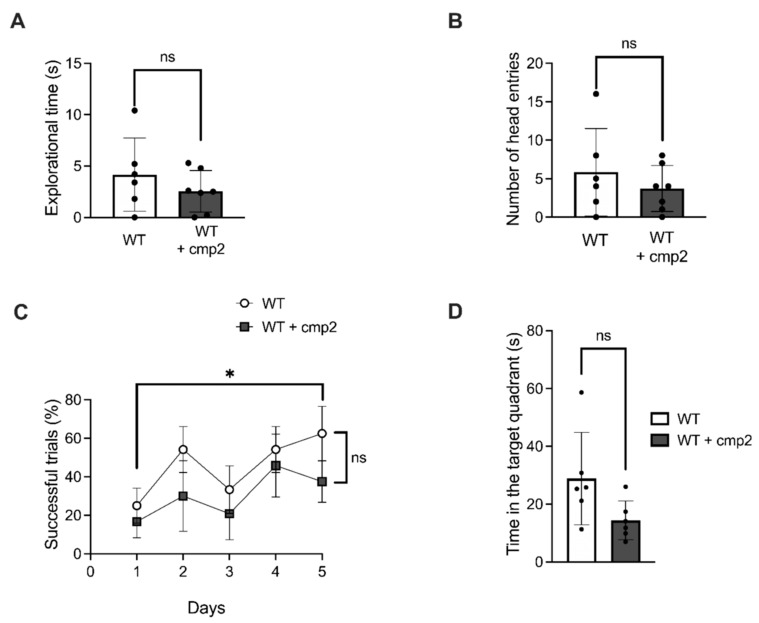
Impact of cmp2 on the recognition memory of WT male mice in the NOR (**A**,**B**) and MWM (**C**,**D**) tests. (**A**) Exploration time of the novel object on the second day. (**B**) Number of head entries in the new object on the second day. The number of male mice tested per group (n): n (WT) = 6, n (WT + cmp2) = 6–7. Data are presented as mean ± SEM, and individual data points are presented as dots. Normal distribution was checked using the Shapiro–Wilk test. Statistical analysis was performed using the Mann–Whitney test or unpaired *t*-test. *: *p* < 0.05, ns: non-significant. Statistical power = 0.52.

**Figure 6 ijms-26-04591-f006:**
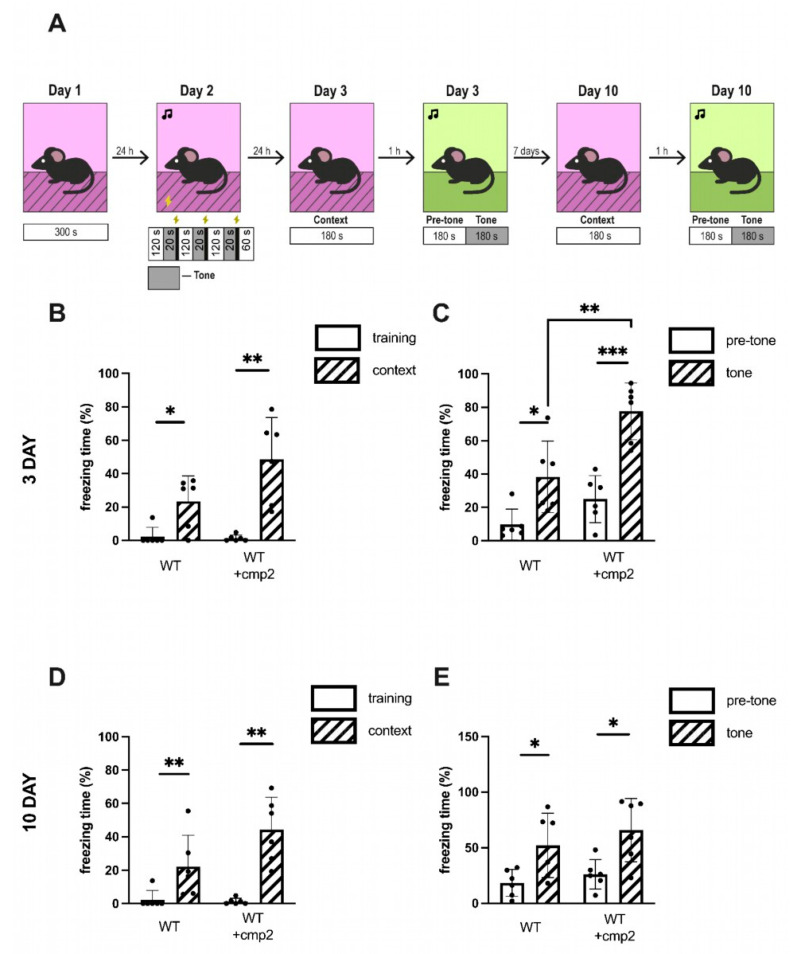
Impact of cmp2 on the context and cued memory of WT mice in the fear conditioning test. (**A**) Schematic illustration of the fear conditioning paradigm. Total freezing percentage during the contextual fear conditioning test of mice performed on day 3 (**B**) and on day 10 (**D**) of the test. Total freezing percentage during the tone fear conditioning test of mice performed on day 3 (**C**) and on day 10 (**E**) of the test. The number of male mice tested per group (n): n (WT) = 6, n (WT + cmp2) = 6. All data are presented as the mean ± SD, and individual data points are presented as dots. Sample distributions were assessed for normality (Shapiro–Wilk test). *p* values indicate significant differences (Mann–Whitney test or *t*-test). ***: *p* < 0.001, **: *p* < 0.01, *: *p* < 0.05. Statistical power ≥ 0.6.

**Figure 7 ijms-26-04591-f007:**
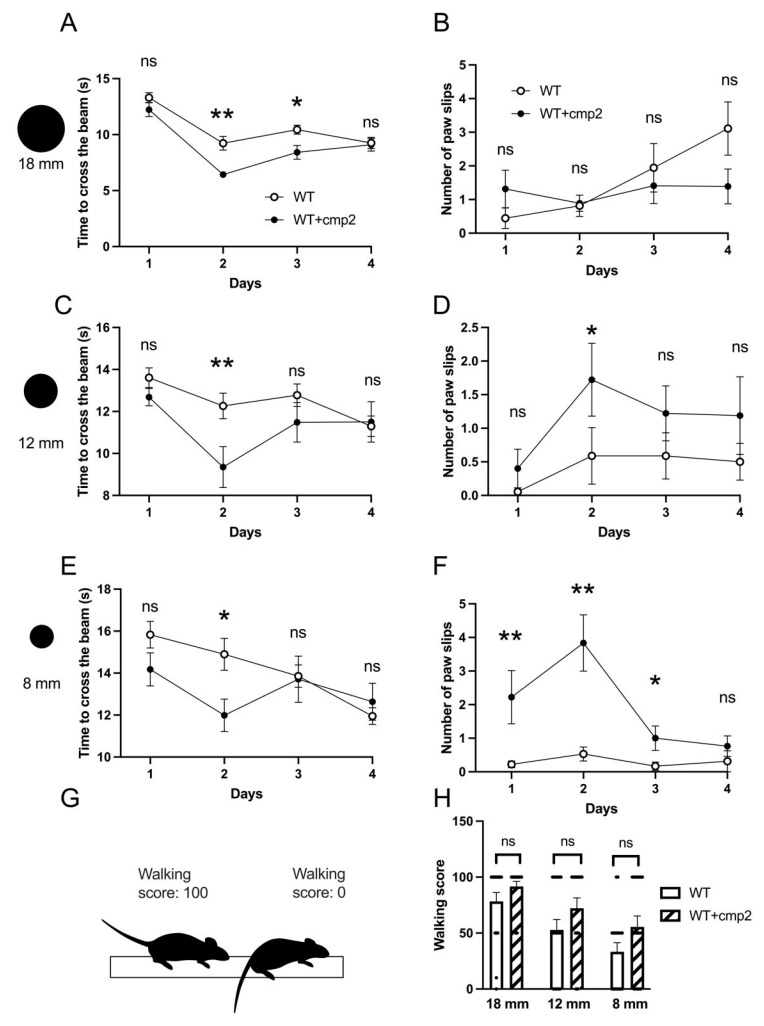
Impact of cmp2 on locomotor parameters of WT male mice in the beam walking test. (**A**,**C**,**E**) Graphs of the average time of beam crossing during 4 days of training on beams of 18 mm, 12 mm, and 8 mm, respectively (each experimental group consists of n = 6 mice, with three attempts for each animal). Statistical difference was measured on the 1st and 4th day. Data are presented as mean ± SEM. (**B**,**D**,**F**) Graphs of the number of times the mice’s paws slipped on beams of 18 mm, 12 mm, and 8 mm, respectively. Data are presented as mean ± SEM. (**G**) Representative picture of walking score estimation. A walking score of «100» is for a normal traverse on all paws. A walking score of «0» is for a “crawling” traverse by dragging hindlimbs. (**H**) Diagram of type of mice movements while beam crossing on day 4. Data are presented as mean ± SEM, and individual data points are presented as dots. Normal distribution was checked using the Shapiro–Wilk test. Statistical analysis was performed using the Mann–Whitney test or unpaired *t*-test. **: *p* < 0.01, *: *p* < 0.05, ns: non-significant. Statistical power ≥ 0.4.

**Figure 8 ijms-26-04591-f008:**
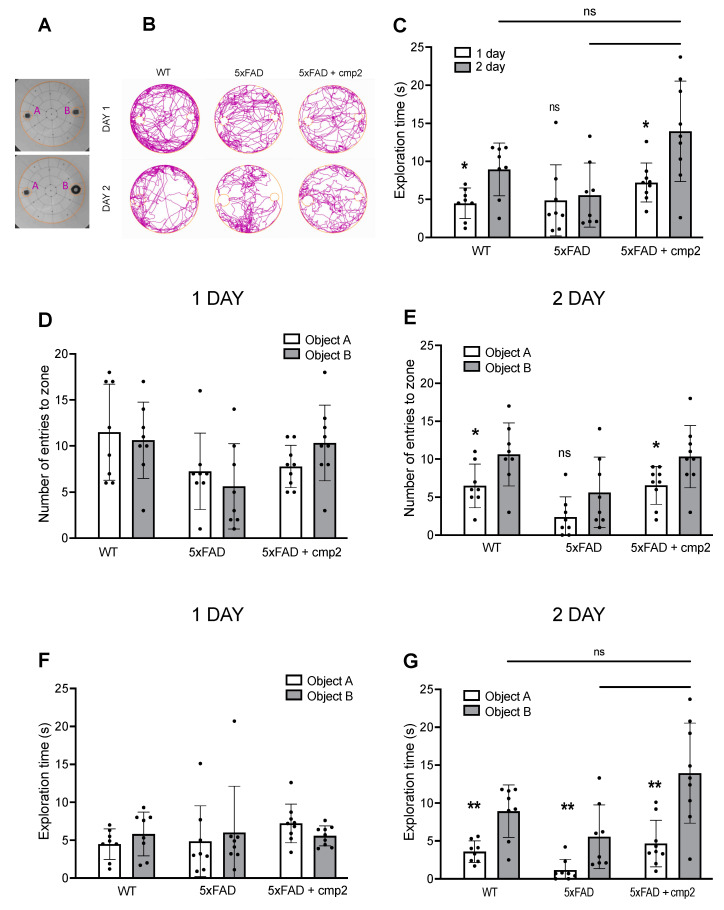
Cmp2 improves the recognition memory of 5xFAD mice in the NOR test. (**A**) The arena with two equal objects (DAY 1) or with the one familiar B and the one novel Object A (DAY 2). (**B**) Representative mice tracks of each experimental group for day 1 and day 2. (**C**) Exploration time of object A. Mice from WT and 5xFAD + cmp2 groups spent more time exploring novel Object A on the second day than on the first. 5xFAD mice spent equal time exploring Object A on the first and second days. On the first day, mice did not prefer objects in terms of number of entries (**D**) or exploring time (**F**). On the second day, the number of entries to zones was significantly different across the WT and 5xFAD + cmp2 groups but not in the 5xFAD group (**E**). The exploring time was significantly different in all groups on the second day (**G**). However, the exploring time of new Object B by mice from the 5xFAD + cmp2 group was significantly higher than in the 5xFAD group. The results are presented as mean with SD, and individual data points are presented as dots. The number of mice tested per group (n): n (WT) = 8, n (5xFAD) = 8, n (5xFAD + cmp2) = 9. Normal distribution was checked using the Shapiro–Wilk test. Statistical analysis was performed using the Mann–Whitney test or *t*-test between two groups and one-way ANOVA following Dunnett’s multiple comparisons test between 5xFAD + cmp2 and the other treatment groups. **: *p* < 0.01, *: *p* < 0.05, ns: non-significant.

**Figure 9 ijms-26-04591-f009:**
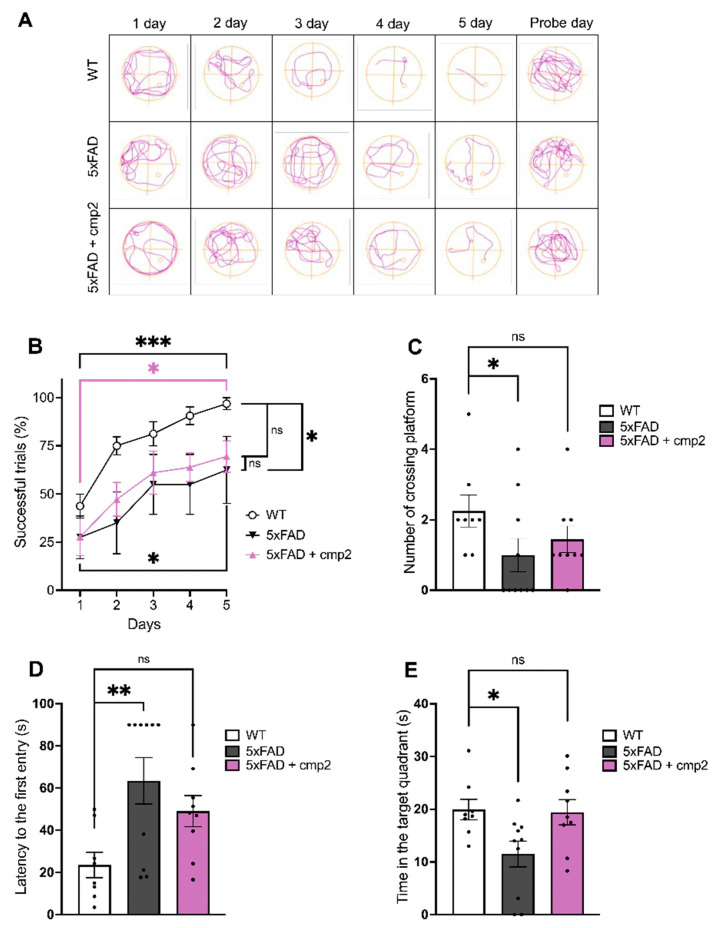
Cmp2 improves the spatial memory of 5xFAD mice in the Morris water maze task. (**A**) Representative swimming traces of each experimental group during the memory test. (**B**) Percentage of successful trials during training days. (**C**) Number of crossings to the platform location. (**D**) Latency of the first entry to the platform location. (**E**) Time spent in the target quadrant on the probe day. All data represent the mean ± SD (**B**) or SEM (**C**,**D**,**E**), and individual data points are presented as dots. The number of mice tested per group (n): n (WT) = 8, n (5xFAD) = 10, n (5xFAD + cmp2) = 9. Normal distribution was checked using the Shapiro–Wilk test. Statistical analysis was performed using the Kruskal–Wallis test with Dunn’s post hoc or one-way ANOVA following Dunnett’s multiple comparisons test. ***: *p* < 0.001, **: *p* < 0.01, *: *p* < 0.05, ns: non-significant.

**Figure 10 ijms-26-04591-f010:**
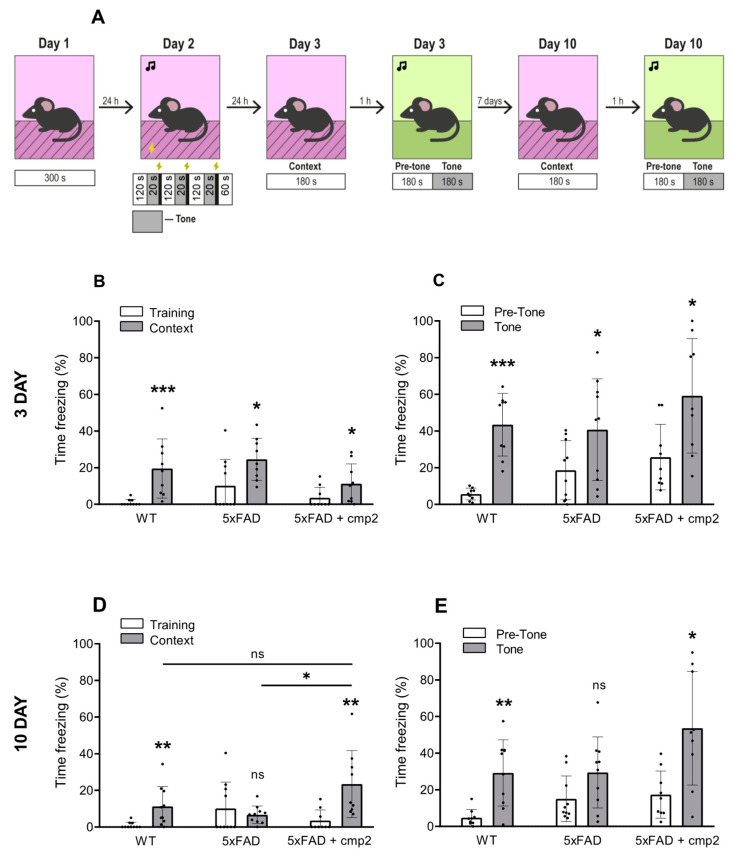
Cmp2 improves the context and cued memory of 5xFAD male mice in the fear conditioning test. (**A**) Schematic illustration of the fear conditioning paradigm. Total freezing percentage during the contextual fear conditioning test of mice performed on day 3 (**B**) and on day 10 (**D**) of the test. Total freezing percentage during the tone fear conditioning test of mice performed on day 3 (**C**) and on day 10 (**E**) of the test. The number of mice tested per group (n): n (WT) = 9, n (5xFAD) = 10, n (5xFAD + cmp2) = 9. All data are presented as the mean ± SD, and individual data points are presented as dots. Sample distributions were assessed for normality (Shapiro–Wilk test) and homogeneity (Bartlett’s test). *p* values indicate significant differences (Mann–Whitney test (**B**,**D**,**E**) or *t*-tests (**C**,**E**)) between conditions (training/context or pre-tone/tone) in the same group or (Kruskal–Wallis test following Dunn’s multiple comparisons test (**D**)) between 5xFAD + cmp2 and the other groups of treatment. ***: *p* < 0.001, **: *p* < 0.01, *: *p* < 0.05, ns: non-significant.

**Figure 11 ijms-26-04591-f011:**
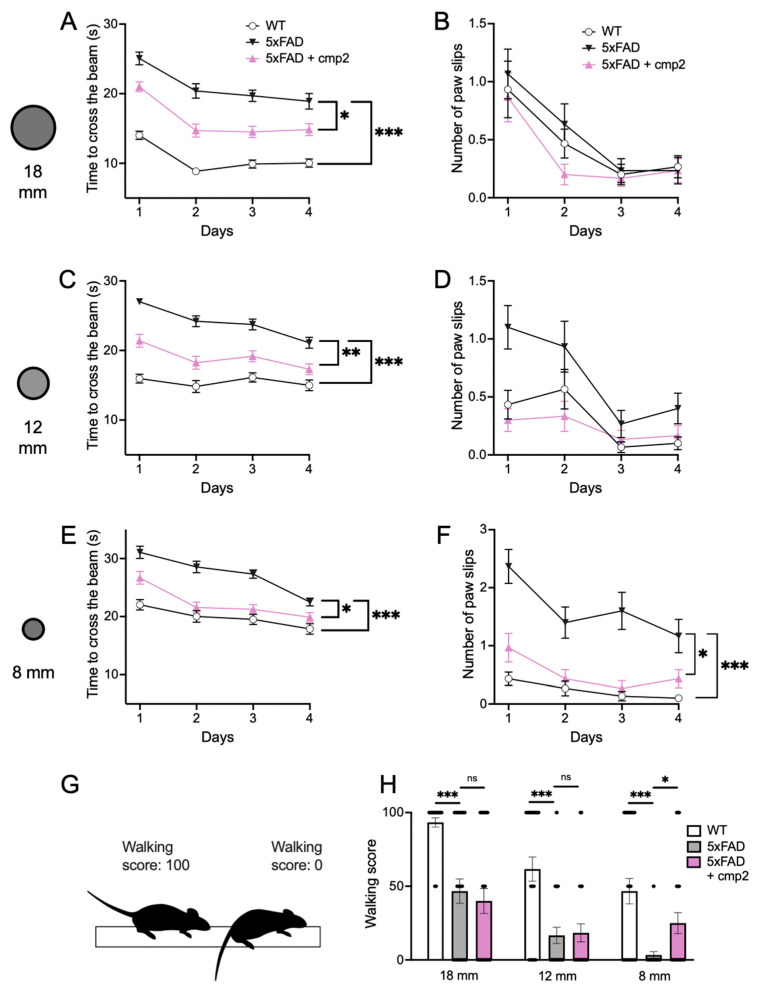
Cmp2 recovers the locomotor parameters of 5xFAD mice in the beam walking test. (**A**,**C**,**E**) Graphs of the average time of beam crossing during 4 days of training on beams of 18 mm, 12 mm, and 8 mm, respectively (each experimental group consists of n = 10 mice and three attempts for each animal). Statistical difference was measured on the 4th day. Data are presented as mean ± SD. (**B**,**D**,**F**) Graphs of the number of times the mice’s paws slipped on beams of 18 mm, 12 mm, and 8 mm, respectively. Data are presented as mean ± SD. (**G**) Representative picture of walking score estimation. A walking score of «100» is for a normal traverse on all paws. A walking score of «0» is for a “crawling” traverse by dragging the hindlimbs. (**H**) Diagram of type of mice movements while beam crossing on day 4. Data are presented as mean ± SEM, and individual data points are presented as dots. Normal distribution was checked using the Shapiro–Wilk test. Statistical analysis was performed using the Kruskal–Wallis test with Dunn’s post hoc or one-way ANOVA following Dunnett’s multiple comparisons test. * *p* < 0.05, ** *p* < 0.005, *** *p* < 0.0001. ns: non-significant.

**Figure 12 ijms-26-04591-f012:**
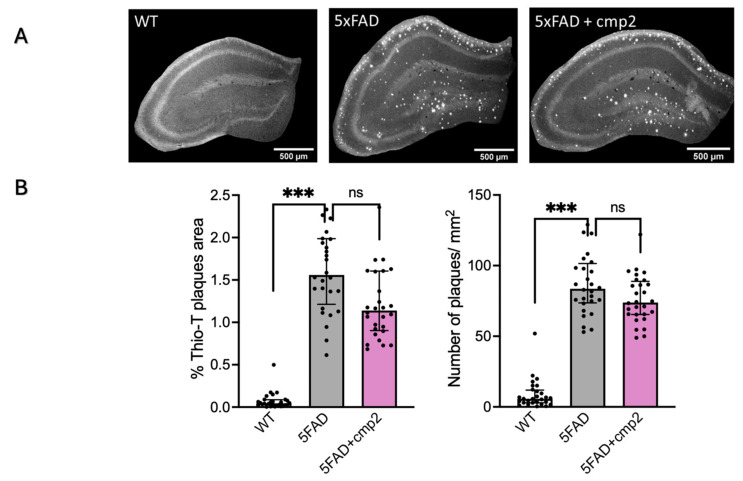
Immunohistochemical staining of hippocampal Aβ-plaques with Thio-T. (**A**) Representative images of hippocampus of 9-month-old mice WT, 5xFAD, 5xFAD + cmp2 with Thio-T staining. (**B**) Quantification of percentages of Thio-T-positive plaques area per hippocampus area and number of Thio-T-positive cells per mm2. Data are presented as median ± Q1/Q3, and individual data points are presented as dots; repeated measurements were taken with the Kruskal–Wallis test and Dunn’s post hoc test. ***: *p* ≤ 0.0001, ns: non-significant, n = 7 mice per group.

**Figure 13 ijms-26-04591-f013:**
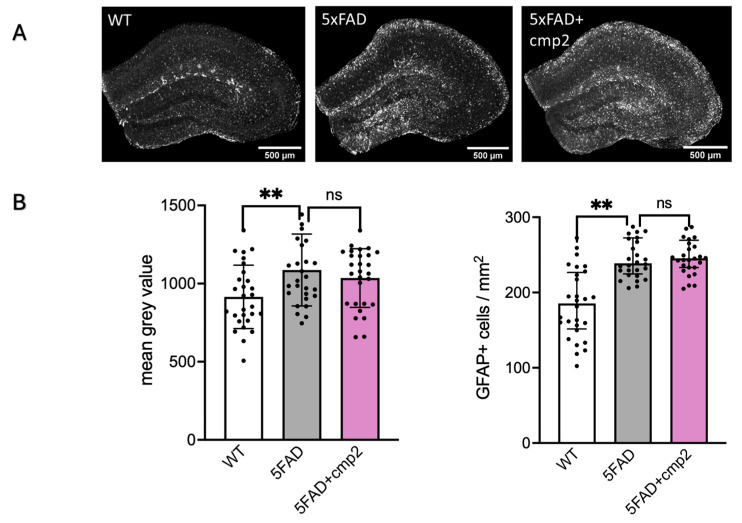
Immunohistochemistry with GFAP antibodies. (**A**) Representative hippocampus of WT, 5xFAD, 5xFAD + cmp2 with GFAP staining at the 9-month mice. (**B**) Estimation of astrogliosis through the analysis of mean grey value of hippocampus area and number of GFAP-positive cells per mm2. Data are represented as mean ± SD, and individual data points are presented as dots; repeated measurements were taken with a one-way ANOVA, Welch’s ANOVA, and Dunnett’s post hoc test. **: *p* ≤ 0.01, ns: non-significant, n = 7 mice per group.

## Data Availability

The datasets used and analyzed during the current study are available from the corresponding author, E. Popugaeva (lena.popugaeva@gmail.com), on reasonable request.

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
