# Peer review of "N-N-Substituted Piperazine, Cmp2, Improves Cognitive and Motor Functions in 5xFAD Mice"

_ijms, 2025, doi:10.3390/ijms26104591_

Round 1

Reviewer 1 Report (Previous Reviewer 1)

Comments and Suggestions for Authors

Now the manuscript looks improved and can be published

Author Response

Reviewer 2 Report (New Reviewer)

Comments and Suggestions for Authors

This manuscript investigates a novel TRPC6-selective activator, cmp2, in the context of Alzheimer’s disease (AD) using the 5xFAD mouse model. Cmp2 is shown to enhance synaptic plasticity and behavioral outcomes. The authors validate its safety profile and acute/chronic toxicity studies. Behavioral tests demonstrate improved memory and motor function in 5xFAD mice treated with cmp2 (10 mg/kg, i.p.), without altering amyloidosis or astrogliosis. These results suggest cmp2 may offer a non-toxic, amyloid-independent approach to AD therapy.

While the study presents compelling preclinical findings, several methodological weaknesses and interpretive gaps must be addressed before publication.

Major Concerns:

  1. The behavioral experiments use n = 6 (WT) and n = 8–10 (5xFAD), which is underpowered for detecting moderate behavioral effects. No power analysis is provided. This undermines the robustness of both positive and negative findings. The authors should include a power calculation or clearly acknowledge this as a limitation.
  2. The study uses female mice for toxicity testing and male mice for behavioral. testing without justification. This inconsistency may introduce sex-dependent variability. The rationale for sex selection should be clarified, and sex should be addressed as a biological variable.
  3. Only a single dose (10 mg/kg) is tested for efficacy. The absence of pharmacokinetics (PK) or dose-response data limits interpretation of both efficacy and specificity. The authors should acknowledge this and recommend future PK/PD profiling and dose optimization.
  4. Cmp2 improves cognition and motor performance in 5xFAD mice but does not reduce amyloid plaque burden or astrogliosis (p.19–20). This raises the question: is cmp2 symptom-relieving or disease-modifying? The discussion should explore potential mechanisms e.g., TRPC6-mediated neuroplasticity, downstream signaling, or cholinergic involvement, to explain these effects in the absence of pathology changes.

Minor Comments:

Line 162 (p.6): “cmp24-month-old” → likely a typo; should be “cmp2, 4-month-old”.

Line 261 (p.10): “ns: non-significant)” → contains an extra parenthesis.

Line 608 (p.23): Sentence breaks mid-way after “kleptose”; appears incomplete or cut off.

Round 2

Reviewer 2 Report (New Reviewer)

Comments and Suggestions for Authors

Thank you for your detailed and thoughtful responses to my comments. I appreciate the clarity and rigor with which you have revised the manuscript. Your additions significantly strengthen the work. I have no further concerns and commend you on a well-executed study.

This manuscript is a resubmission of an earlier submission. The following is a list of the peer review reports and author responses from that submission.

Round 1

Reviewer 1 Report

Comments and Suggestions for Authors

This manuscript deals with an interesting and important topic, but I have doubts about the interpretation of some of the data presented.

1. The main doubts are related to the form of the control/5xFAD data analysis. As far as I could understand, in no case was the effect of cmp2 shown on the control group. In my opinion, this negates the specificity of the effect of cmp2 on the 5xFAD mutants, versus the impact on behavior as such.

2. It is unclear why control mice show better results in the Morris water maze on the first day. Is it a form of intuition?

3. The same applies to the locomotor test (Fig. 8).

Author Response

Reviewer 1

This manuscript deals with an interesting and important topic, but I have doubts about the interpretation of some of the data presented.

Reviewer 1, question 1. The main doubts are related to the form of the control/5xFAD data analysis. As far as I could understand, in no case was the effect of cmp2 shown on the control group. In my opinion, this negates the specificity of the effect of cmp2 on the 5xFAD mutants, versus the impact on behavior as such.

Author’s response:

We thank Reviewer 1 for positive evaluation of the manuscript and providing constructive critics. We agree that including a WT + cmp2 group would have been beneficial for this study. However, cmp2 effect on LTP induction in WT group as well as influence on spines were previously investigated (Zernov et al 2024, Sci Rep). No effects of cmp2 on LTP induction in WT group as well as on spines formation were observed (Fig 5, Zernov et al 2024, Sci Rep) thus we skipped WT+cmp2 group from behavioral tests. In addition, in order to respect animal ethical issues, we tried to keep the number of mice used per experiment as small as possible.

Reviewer 1, question 2. It is unclear why control mice show better results in the Morris water maze on the first day. Is it a form of intuition?

Author’s response:

We draw Reviewer 1 attention to that the difference in escape latency between WT and 5xFAD mice on day 1 in not statistically significant. Nevertheless, the age of mice tested in MWM is around 8 months. By this age 5xFAD mice are dramatically differ from WT littermates in behavioral indicators. Similar differences in spatial memory between WT and 5xFAD mice at day 1 during spatial learning training was previously reported by other research groups (https://doi.org/10.3390/biomedicines11020599, https://doi.org/10.4062/biomolther.2019.046).

Reviewer 1, question 3. The same applies to the locomotor test (Fig. 8).

Author’s response:

5xFAD mice do perform worse than WT littermates in locomotor learning test even at day 1 depending on the age (older 5xFAD mice show more pronounced difference to WT) as confirmed by other research group (PMID: 30426678, DOI: 10.1111/gbb.12538). Therefore, data provided by our research group on Fig 8 is consistent with literature data.

Reviewer 2 Report

Comments and Suggestions for Authors

This article reports the effect of a piperazine derivative on several cognitive and motor functions on an Alzheimer model mouse. The compound used (cmp2) it is a piperazine derivative described by the same group in a previous article (Zernov et al, 2024) having a neuroprotective effect by selectively activating TRPC6 channels (while not affecting TRPC3 nor TRPC7). The authors select this compound among others by using an online tool: Prediction of Activity Spectra for Substances [(PASS); http://www.way2drug.com/passonline] (Zernov et al, 2024).

In the present study the authors report a recovery of several cognitive properties (recognition memory, spatial memory, recognition and cued fear memory and motor skills) in the 5xFAD mouse while also doing a preliminary characterization of the safety profile of the compound including mutagenic potential and effect on body weight. According to the authors the effect of cmp2 is due to its neuroprotective properties by activating TRPC6 and it does not seem to affect amyloidosis nor astrogliosis.

The article is well written in general and it describes the cmp2 effects on the chosen experiments very clear. Nevertheless, I have a major objection to the conclusion that I think it needs to be addressed:

Piperazine derivatives are a major source of pharmacological active compounds and some of them have effects on the cholinergic system (KaraytuÄŸ MO, 2023). Pharmacological compounds, with effect on the cholinergic system, have been used to reduce AD symptoms and they improve the performance in the same or similar tasks like the ones used in this article (Pepeu et al, 2018; Janas et al, 2005; Dong et al, 2005; Pant et al, 2024). I believe the authors should show that cmp2 does not have an effect on the cholinergic system (i.e. acetylcholinesterase) that might also explain the behavioral results. I tried myself to use the above-mentioned online tool to study cmp2 interaction with the acetylcholinesterase but I was not able to register in the web. I believe that addressing this point will clarify and strengthen the authors claim about the cmp2 effect.

Minor points:

-          Please include individual data points in all the bar plots.

-          Figure 6 is labelled as Figure 1 in the legend, please correct.

-          In Fig 6B WT are missing the error bars.

Author Response

Reviewer 2

Piperazine derivatives are a major source of pharmacological active compounds and some of them have effects on the cholinergic system (KaraytuÄŸ MO, 2023). Pharmacological compounds, with effect on the cholinergic system, have been used to reduce AD symptoms and they improve the performance in the same or similar tasks like the ones used in this article (Pepeu et al, 2018; Janas et al, 2005; Dong et al, 2005; Pant et al, 2024). I believe the authors should show that cmp2 does not have an effect on the cholinergic system (i.e. acetylcholinesterase) that might also explain the behavioral results. I tried myself to use the above-mentioned online tool to study cmp2 interaction with the acetylcholinesterase but I was not able to register in the web. I believe that addressing this point will clarify and strengthen the authors claim about the cmp2 effect.

 Author’s response:

We thank Reviewer 2 for careful reading of the manuscript and pointing to the important issue that we have forgotten to mention in the manuscript. Changes made in the text are highlighted by red color.

We have previously found out that cmp2 might inhibit both acetylcholinesterase and butyrylcholinesterase [34], please refer to the manuscript]. However, these are only in silico data. In order to prove cmp2 as an effective inhibitor of cholinergic enzymes in vitro and in vivo data are needed.

We have added following paragraph to the discussion section (lines 440-449):

Important to note that piperazine derivatives have been shown to inhibit both acetylcholinesterase (AСhE) and butyrylcholinesterase (BuChE)[30] . Pharmacological compounds, with effect on the cholinergic system, have been used to reduce AD symptoms and improve the performance in the similar behavioral tasks like the ones used in the current article [31–33]. Indeed, our previous in silico data demonstrate influence of cmp2 on cholinergic system [34]. Thus, the likelihood of cmp2 positive impact on 5xFAD animal behavior via modulating TRPC6 and cholinergic -dependent signaling pathways is high. However, cmp2 effect on the cholinergic system is confirmed only in silico. Further in vitro and in vivo experiments are needed to validate cmp2 as effective AChE and BuChE inhibitor.

Reviewer 2, Minor points:

Reviewer 2, point 1 Please include individual data points in all the bar plots.

Author’s response:

We have added individual data points in Figures 1 to 8. Please see changed figures in revised version of the manuscript.

Reviewer 2, point 2 Figure 6 is labelled as Figure 1 in the legend, please correct.

Author’s response:

We are sorry for the misprint. We have corrected the figure’s number in a revised version of the manuscript.

Reviewer 2, point 3 In Fig 6B WT are missing the error bars. 

Author’s response:

We are sorry for missing error bars. We have fixed Fig 6B in revised version of the manuscript.

Round 2

Reviewer 1 Report

Comments and Suggestions for Authors

Despite the explanations given by the authors, I continue to consider the lack of effect of the activator on the control group a serious omission.

One can have different attitudes to the use of the 5xFAD line. For example, they can be considered as animals with cognitive or motor deficits, outside the specific context of Alzheimer's disease of the genetic (familial) type. However, this approach gives rise to a series of questions, the main one being: why was this model chosen? There are other models and pharmacological conditions with varying degrees of cognitive and / or motor impairment. Will the agonist (activator) act there as well? Moreover, (as the authors themselves note in the answers to my questions 2 and 3) 5xFAD mice show significant motor deviations. Is this good for the experiment? Of course not, because Cmp2 can restore motor, not cognitive dysfunction, making 5xFAD more mobile. Whether this is true remains unclear.

However, the authors clearly did not mean to use 5xFAD simply as a convenient but not fundamental example of dysfunctions, as evidenced by the reference to Alzheimer's disease in the abstract. The authors' ambitions are precisely that they have discovered a means of targeting this condition (and without affecting amyloid plaques). I have no reason to deny the possibility of such an achievement, but I cannot take it on faith without the most basic control. Once again: in my opinion, a thorough study of the effect of Cmp2 on control mice is fundamental.

Reviewer 2 Report

Comments and Suggestions for Authors

In my opinion, the fact that cmp2 might have potential effects on the cholinergic system requires a change in the article beyond rephrasing the paragraph I wrote in the discussion.

The way the article is written at the moment gives the impression that activating TRPC6 can lead to improvement in AD symptoms in the mouse model but, as I stated before, this effect might be explained solely by an effect on the cholinergic system (i.e. inhibition of AchE). Since the possibility of interaction of cmp2 with AchE is a potential effect known and acknowleged by the authors (they even added a publication from their own group that was not included in the original submission) this should be reflected in the introduction and not only in the discussion.

Overall I think the article is interesting and deserves to be published once this point is adressed.

Round 3

Reviewer 1 Report

Comments and Suggestions for Authors

Unfortunately, I do not see any significant improvement, especially since they cannot be made in three days

Reviewer 2 Report

Comments and Suggestions for Authors

My concerns are adressed